# Ref-Diff: Zero-shot Referring Image Segmentation with Generative Models

## Abstract

Zero-shot referring image segmentation (RIS) presents a significant challenge. It requires identifying an instance segmentation mask using referring descriptions, without having been trained on such paired data. While existing zero-shot RIS methods mainly utilize pre-trained discriminative models (*e.g.*, CLIP), this study observes that generative models (*e.g.*, Stable Diffusion) can discern relationships between various visual elements and text descriptions—an area yet to be explored in this task. In this work, we introduce the **Ref**erring **Diff**usional Segmentor (Ref-Diff), a model that leverages the fine-grained multi-modal information derived from generative models. Our findings show that even without any external proposal generator, our Ref-Diff with a sole generative model outperforms SOTA weakly-supervised models on RefCOCO+ and RefCOCOg. Notably, when combining both generative and discriminative models, our Ref-Diff+ surpasses competing methods by a substantial margin. This highlights the constructive role of generative models in this domain, providing complementary advantages alongside discriminative models to enhance referring segmentation. Our source code will be publicly available.

## 1    Introduction

Referring Image Segmentation (RIS) aims to find the instance region that aligns semantically with a given textual description. Unlike semantic segmentation, this task requires distinguishing instances of the same class, *e.g.*, the tallest boy among these four children. However, annotating such types of pairs (*i.e.*, image, text description, and ground-truth instance mask) is costly and time-consuming. A recent weakly-supervised RIS approach (Strudel et al., 2022) endeavors to alleviate the annotation difficulties, but it still needs specific pairs of images and referring texts for training. In comparison, a zero-shot solution is more valuable but might amplify the challenges further. On the one hand, it is training-free, and no referring annotation is required. On the other hand, it needs a deeper understanding of the relationship between text and the visual elements.

Recent multi-modal pre-training models have shown impressive abilities in vision and language understanding. As one of the most representative discriminative models among them, CLIP (Radford et al., 2021), explicitly learns the global similarity between image and text through contrastive learning. This model has been pivotal in enhancing performance across various tasks, such as object detection (Zhong et al., 2022), image retrieval (Radford et al., 2021), and semantic segmentation (Dong et al., 2023). However, directly applying a model like CLIP to zero-shot RIS is impractical, as it is optimized to capture the global similarity of text and images, which cannot learn the intricate visual elements corresponding to a referring text. To mitigate this, Yu et al. (2023) propose a global and local CLIP to bridge the gap between discriminative models and pixel-level dense prediction. Nevertheless, we observe that discriminative models themselves struggle to localize visual elements accurately. Concurrently, generative models like Stable Diffusion (Rombach et al., 2022), DALL-E 2 (Ramesh et al., 2022), and Imagen (Saharia et al., 2022) are attracting great attention due to their ability to synthesize realistic or imaginative images. The semantically coherent output demonstrates that they have implicitly captured the relationships between various visual elements and texts. Nonetheless, in contrast to discriminative models, generative models are seldom exploited in zero-shot RIS, despite their potential in capturing intricate text-visual correlations.

In this work, we explore the potential benefits of using generative models for the zero-shot RIS task. To this end, we introduce a novel Referring Diffusional Segmentor (REF-DIFF). It leverages fine-grained multi-modal information inherent in generative models to exploit the relationship between referring expressions and various visual elements within the image. Besides, previous works usually adopt CLIP to rank the proposals from an external proposal generator (Kirillov et al., 2023). In contrast, our REF-DIFF can generate these instance proposals via the generative models it utilizes. This suggests that our REF-DIFF can operate independently without external proposal generators, demonstrating its self-sufficiency within the generative model framework. Moreover, we analyze the merits of discriminative and generative models and integrate them into a unified RIS framework.

Experiments show that without the use of any external proposal generator, the generative model in our REF-DIFF outperforms the SOTA weakly-supervised methods on REFCOCO+ and REF-COCOG. Notably, when incorporating an external proposal generator and a discriminative model, our REF-DIFF+ significantly outperforms all the competing methods. Comprehensive quantitative and qualitative assessments validate the values of the generative model for this task.

The main contributions can be summarized as follows:

- We show that generative models can be utilized to improve the zero-shot RIS performance by exploiting the implicit relationships between visual elements and text descriptions.

- We propose REF-DIFF, a model without external proposal generators and discriminative models, achieves performance comparable to weakly-supervised SOTA methods.

- By effectively combining generative and discriminative models in a unified framework, our REF-DIFF+ leverages the strengths of each to achieve superior RIS outcomes.

## 2 RELATED WORK

**Zero-shot Referring Segmentation.** Referring image segmentation is one of the most fundamental and challenging tasks, as it involves a fine-grained understanding of both vision and language. Following a fully-supervised formulation, previous works (Xu et al., 2023; Yan et al., 2023; Yang et al., 2023; Liu et al., 2023; Ding et al., 2022; Yang et al., 2022; Zhao et al., 2023) require labor-intensive training annotations, *i.e.*, referring expressions and pixel-level masks. However, due to the absence of large-scale training annotations, these methods are often limited in their scalability and out-of-domain samples (Strudel et al., 2022). With the remarkable progress of discriminative vision-language pre-training (Radford et al., 2021), recent works (Strudel et al., 2022; Yu et al., 2023) explore their open-vocabulary recognition in weakly or zero-shot RIS. Despite their considerable performance, discriminative models that learn the global similarity of text and image have inherent limitations in deeply understanding the object delineation or the fine-grained relationships between visual elements and text description. In contrast, our REF-DIFF utilizes the fine-grained understanding through generative models and thereby obtains more accurate predictions.

**Visual Generative Models for Non-Synthesized Tasks.** Large-scale text-to-image generative models (Rombach et al., 2022; Ramesh et al., 2022; Saharia et al., 2022) have achieved tremendous progress in imaginary generation and creative applications (Wei et al., 2023; Ruiz et al., 2023; Gal et al., 2023; Zhang & Agrawala, 2023; Mou et al., 2023). Apart from generation-related tasks, these generative models also demonstrate preeminent capabilities in fine-grained image understanding, *e.g.*, semantic segmentation (Zhao et al., 2023; Wu et al., 2023; Asiedu et al., 2022; Baranchuk et al., 2022; Amit et al., 2021), object detection (Cheng et al., 2023; Chen et al., 2022), dense prediction (Ji et al., 2023; Saxena et al., 2023), and classification (Li et al., 2023; Clark & Jaini, 2023). Therefore, most existing research focuses on transfer learning or constructing synthetic data for specific tasks. As generative models have shown a powerful ability to understand the text description and generate corresponding image content, a few studies have begun to explore the application of using generative models in zero-shot image classification. However, the performance of generative models in zero-shot RIS tasks remains underexplored.

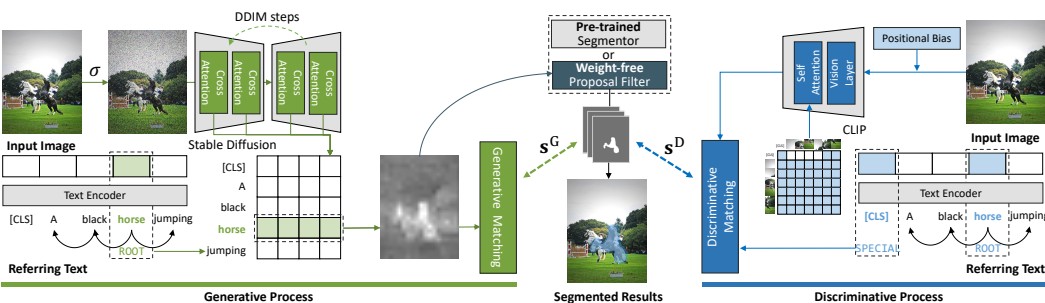

Figure 1: **Overview of our REF-DIFF and REF-DIFF+**. **Left**: Our proposed generative process generates a correlation matrix between the referring text and the input image. This matrix serves as an alternative weight-free proposal generator and generative segmentation candidates $\mathbf{s}^{\mathrm{G}}$. **Right**: The discriminative process generates the discriminative candidates $\mathbf{s}^{\mathrm{D}}$. The final RIS result can be obtained either from the generative candidates or a combination of both generative and discriminative candidates. Note that our REF-DIFF refers to the model that adopts the generative process and uses our weight-free proposal generator, whereas REF-DIFF+ integrates both the generative and discriminative processes and operates in conjunction with the pre-trained segmentor.

## 3 REF-DIFF AND THE EXTENSION TO REF-DIFF+

### 3.1 PROBLEM FORMULATION AND INFERENCE PIPELINE

Given an image $\mathbf{x} \in \mathbb{R}^{W \times H \times C}$ and a referring text $T$, Referring Segmentation aims to output a segmenting mask $\mathbf{m} \in \{0,1\}^{W \times H}$ indicating the referring regions of the text $T$ in image $\mathbf{x}$. Here $W$, $H$, and $C$ represent the width, height, and channel of the image, respectively. In the Zero-shot Referring Segmentation settings, the model cannot access any training data of Referring Segmentation, including images, referring texts, and instance mask annotations.

Our proposed framework is depicted in Figure 1. Given an image and the referring text, our REF-DIFF generates a correlation matrix using the Generative Process, which can be used as 1) an alternative weight-free proposal generator and 2) a set of referring segmentation candidates. Optionally, our REF-DIFF can integrate the discriminative model with our proposed generative model within a unified framework (*i.e.*, REF-DIFF+). The final similarity for each mask proposal is obtained as:

$$\mathbf{s}_i = \alpha \mathbf{s}_i^{\mathrm{G}} + (1 - \alpha)\mathbf{s}_i^{\mathrm{D}}, \tag{1}$$

where $\alpha$ is a hyper-parameter. $\mathbf{s}_i^{\mathrm{G}}$ and $\mathbf{s}_i^{\mathrm{D}}$ are the generative and discriminative scores between the referring text and the $i$-th proposal, respectively. The final referring segmentation result is determined by selecting the proposal with the highest similarity score:

$$\hat{\mathbf{m}} = \arg\max_{\mathcal{M}_i} \mathbf{s}_i. \tag{2}$$

In REF-DIFF, $\alpha$ is set to 1, where only the generative model is adopted for this task and the proposals are obtained from our weight-free proposal filter. In REF-DIFF+, $\alpha$ is set to a number between 0 and 1, and proposals are obtained from external pre-trained segmentor to obtain a better performance. In the following sections, we provide a detailed introduction to each module of our framework.

### 3.2 GENERATIVE PROCESS

STABLE DIFFUSION (Rombach et al., 2022) is an effective generative model that consists of a series of inverse diffusion steps to gradually transform random Gaussian noise into an image. Therefore, it cannot directly operate on a real image to obtain its latent representations. Fortunately, as the diffusion process is computable, we can take the real image as one intermediate state generated by a diffusion model and run it back to any step of the generation. In this work, we add a specific amount of Gaussian noise to obtain $\mathbf{x}_t$ and then continue this process without compromising information:

$$\mathbf{x_t} = \sigma_t(\mathbf{x}), \tag{3}$$

where $\sigma_t$ is the function to obtain the noised image in step $t$ and $t$ is a hyper-parameter.

During the inverse diffusion process, let $\Psi_{\mathrm{lan}}$ and $\Psi_{\mathrm{vis}}$ be the text and image encoder of the generative model, respectively. In the generative model, the referring text $T$ is encoded into text features

using $\mathbf{K} = \Psi_{\text{lan}}(T) \in \mathbb{R}^{l \times d}$, where $l$ is the token number and $d$ is the dimension size for latent projection. Similarly, for the $i$-th step, the visual image $\mathbf{x_i}$ is projected to image features using $\mathbf{Q} = \Psi_{\text{vis}}(\mathbf{x_i}) \in \mathbb{R}^{w \times h \times d}$. Here, $w$ and $h$ are the width and height of encoded image features. The cross-attention between the text and image features can be formulated as:

$$\mathbf{a} = \text{Softmax}(\frac{\mathbf{Q}\mathbf{K}^{\top}}{\sqrt{d}}) , \tag{4}$$

where $\mathbf{a} \in \mathbb{R}^{w \times h \times l \times N}$, and $N$ is the number of attention heads. Following (Wu et al., 2023), we obtain the overall cross-attentions by averaging the value of each attention head to $\bar{\mathbf{a}} \in \mathbb{R}^{w \times h \times l}$. The cross-attention matrix $\bar{\mathbf{a}}$ represents the correlation detected between each token in the referring text and each region feature in the image. In general, a higher value in $\bar{\mathbf{a}}$ indicates a better correlation between the token and region features. This can be used to locate the related referring regions.

In the inverse diffusion process, the generative model captures the overall semantics of the language condition. However, the corresponding attention region for each token is not necessarily the same as they have different semantic representations (see Sec. 4.6). Without loss of generality, a referring text $T$ is a sentence that describes the characteristics of a specific instance. To obtain the preferred token from the whole text description, we use syntax analysis to obtain its root token (*i.e.*, the ROOT element in the syntax tree). Generally, the ROOT token in the latent space captures the contextual correlations (*e.g.*, the token "horse" in Figure 1 contains contextural representations from "black" and "jumping"). Then, the attention region projected for this root token has a higher probability of being the referring region. Let $k$ be the index of the root token, and let $\bar{\mathbf{a}}_k \in \mathbb{R}^{w \times h}$ denote the cross-attention matrix of the root token. We normalize and resize this cross-attention matrix by:

$$\mathbf{c} = \phi_{w \times h \to W \times H}\left(\frac{\bar{\mathbf{a}}_k - \min(\bar{\mathbf{a}}_k)}{\max(\bar{\mathbf{a}}_k) - \min(\bar{\mathbf{a}}_k) + \epsilon}\right) , \tag{5}$$

where $\epsilon$ is a small constant value. Here, $\phi_{w \times h \to W \times H}$ is a bi-linear interpolation function used to resize the attention map to the same resolution as the given image.

## 3.3 Discriminative Process

During the image encoding process by the discriminative model CLIP, the spatial position is inevitably attenuated. We observe that referring text descriptions usually contain explicit direction clues (*e.g.*, left, right, top, and bottom), which are valuable but have been ignored in previous works. To emphasize such types of positional information, we propose a positional bias to explicitly encode the image with the given direction clues. This is achieved through element-wise multiplication $\mathbf{x} \odot \mathbf{P}$, where $\mathbf{P} \in \mathbb{R}^{W \times H \times C}$ is a positional bias matrix. Specifically, if the text, after syntactic analysis, contains explicit direction clues, $\mathbf{P}$ will be a soft mask with values ranging from 1 to 0 along the given direction axis. Lower values indicate regions that should receive less attention. Conversely, if no direction clue is detected, $\mathbf{P}$ will be a matrix filled with 1.

Finally, the ultimate representation $\mathbf{v}_i \in \mathbb{R}^d$ for each proposal $\mathcal{M}_i$ in discriminative process is:

$$\mathbf{v}_i = \beta f_{\mathcal{M}_i}(\mathbf{x} \odot \mathbf{P}) + (1 - \beta)f(\mathbf{x} \odot \mathcal{M}_i), \tag{6}$$

where $f$ and $f_{\mathcal{M}_i}$ are the vanilla CLIP image encoder and CLIP image encoder with modified self-attention based on mask proposal $\mathcal{M}_i$. Since the discriminative model (*i.e.*, CLIP) is expected to encode the instance within each proposal region $\mathcal{M}_i$ while disregarding other regions for reducing disturbances. To achieve this, we assign a weight of 0 to the attention values between the [CLS] token and the patch tokens outside the current proposal $\mathcal{M}_i$. In this work, we utilize the output of the penultimate layer as the final representation, which is inspired by the observation that the representation in the last layer encapsulates the information of the entire image rather than the specific proposal region of interest.

## 3.4 Proposals Extracting and Matching

**Weight-free Proposal Filter.** Since the generative models inherently encode instance representations, we can derive proposals from their cross-attention matrix $\mathbf{c}$. In this work, we introduce a weight-free proposal filter to generate a series of mask proposals. This is formulated as:

$$\mathcal{M} = \{\psi(\mathbf{c} \geq \mu) | \mu \in \{5\%, 10\%, ..., 95\%\}\} , \tag{7}$$

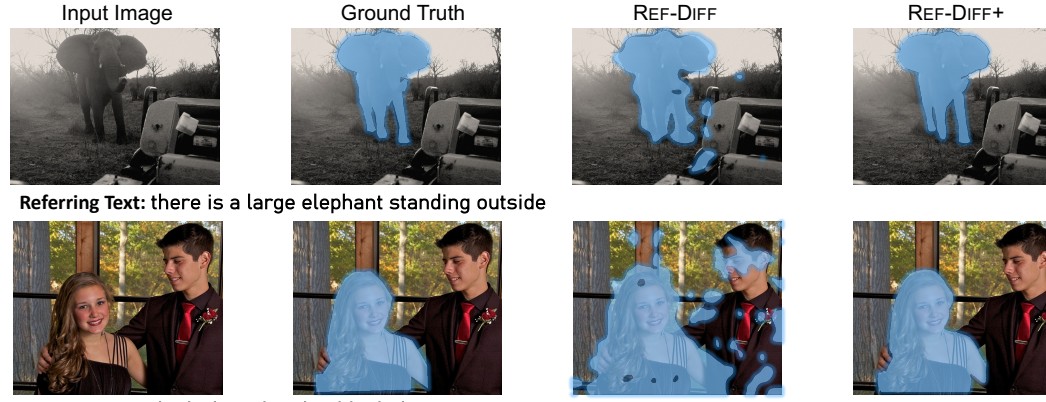

| Input Image | Ground Truth | REF-DIFF | REF-DIFF+ |
| --- | --- | --- | --- |

**Referring Text:** there is a large elephant standing outside

**Referring Text:** the little girl with a black dress

Figure 2: **Effectiveness of generative model in segmentation capability.** REF-DIFF is capable of segmenting the right content even without the assistance of the pre-trained segmentor and CLIP. Combined with the pre-trained segmentor, REF-DIFF+ achieves precise segmentation of the expected regions.

where $\psi$ is a binarization function with a given predefined threshold value $\mu$. Different from other works (Yu et al., 2023) which rely on external proposal generators and CLIP filters, the generative models in this work can efficiently and effectively produce the expected proposals. This approach offers a streamlined solution for obtaining high-quality proposals without additional tools.

**Pre-trained Segmentor.** If a reliable segmentor is available, we can also obtain proposals from it in a flexible manner. By leveraging the capability of generative and discriminative models for semantic understanding, we can refine the proposals and prioritize those that align closely with the given referring description. This combined approach of using a segmentor for initial proposal generation ensures that the proposals are coherent and better aligned with the given referring expression.

**Generative Matching.** After obtaining proposals either from the weight-free proposal filter or the pre-trained segmentor, the next step is to find the most similar proposal based on the cross-attention matrix. In this work, we quantify the similarity between the given referring text and all the proposals by measuring the distance on cross-attention matrix $\mathbf{c}$ and $\mathcal{M}_i$:

$$\mathbf{s}_i^{\mathrm{G}} = \frac{|\mathbf{c} \odot \mathcal{M}_i|}{|\mathcal{M}_i|} - \frac{|\mathbf{c} \odot (1 - \mathcal{M}_i)|}{|1 - \mathcal{M}_i|}. \tag{8}$$

**Discriminative Matching.** Given a referring text, we obtain the mean representation $\mathbf{r} \in \mathbb{R}^d$ of the global text and the local subject token using the CLIP text encoder, which serves as the features of the referring text. To find out the most probable proposal from the perspective of the discriminative model, we calculate the similarity between the features of referring text $\mathbf{r}$ and visual representation $\mathbf{v}_i$ of each mask proposal $\mathcal{M}_i$, which is defined as:

$$\mathbf{s}_i^{\mathrm{D}} = \mathbf{v}_i \mathbf{r}^\top. \tag{9}$$

This similarity allows us to identify the proposal that best aligns with the referring text. Higher similarity scores indicate a stronger correspondence between the proposal and the text, indicating a higher likelihood of being the correct segmentation result.

## 4 EXPERIMENTS

### 4.1 IMPLEMENTATION DETAILS

Our REF-DIFF is a zero-shot solution, so we only need an inference process, without any training images and annotations. All experiments are conducted on a Tesla A100 GPU. We use the pre-trained STABLE DIFFUSION (Rombach et al., 2022) (V1.5) as our generative model. Since existing works mainly focus on using pre-trained segmentor and discriminative model (Radford et al., 2021), for a fair comparison, we select SAM (Kirillov et al., 2023) as the external segmentor and adopt CLIP of VIT-B/16 as the discriminative model. We set $t$, $\alpha$, and $\beta$ to 2, 0.1, and 0.3, respectively.

## 4.2 EXPERIMENTAL SETUP

Following (Yu et al., 2023; Strudel et al., 2022), we adopt mIoU and oIoU as the evaluation metrics and apply them to three widely-used benchmarks, including RefCOCO (Nagaraja et al., 2016), RefCOCO+ (Nagaraja et al., 2016), and RefCOCOg (Mao et al., 2016). For fairly comparing with existing works, we conduct the experiments under two settings: a) Zero-shot RIS without a pre-trained segmentor and CLIP. b) Zero-shot RIS using a pre-trained external segmentor and CLIP; The first setting allows us to analyze the effectiveness of the generative model in this task and the latter one will prove the effectiveness of the collaboration of discriminative and generative models.

For setting a), we choose the weakly-supervised method TSEG (Strudel et al., 2022) as the baseline model. It is not open-sourced and only provides the validation results of mIoU, so we did not report its results on other settings and test sets. In this setting, our REF-DIFF uses a sole generative model without proposal generator (*e.g.*, SAM) and CLIP.

For setting b), we select five competing baselines. 1)~3) Three zero-shot baselines from (Yu et al., 2023), including REGION TOKEN, CROPPING, and GLOBAL-LOCAL CLIP. 4)~5) Prior SOTA method adopted with SAM segmentor and a new zero-shot baseline SAM-CLIP proposed by us, considering the remarkable performance of SAM (Kirillov et al., 2023) in segmentation. In this setting, our REF-DIFF+ combines both generative and discriminative models and uses the proposal generator as other methods.

Table 1: **The oIoU comparison.** The improvement is statistically significant with $p < 0.01$ under $t$-test. † denotes weakly-supervised method trained on datasets.

| METHODS | REFCOCO | | | REFCOCO+ | | | REFCOCOG | | |
|---|---|---|---|---|---|---|---|---|---|
| | VAL | TESTA | TESTB | VAL | TESTA | TESTB | VALU | TESTU | VALG |
| *Without Pre-trained Segmentor* | | | | | | | | | |
| TSEG† | - | - | - | - | - | - | - | - | - |
| **REF-DIFF** | **16.12** | **14.10** | **15.73** | **17.50** | **14.84** | **16.48** | **20.39** | **16.82** | **16.92** |
| *With Pre-trained Segmentor* | | | | | | | | | |
| REGION TOKEN | 21.71 | 20.31 | 22.63 | 22.61 | 20.91 | 23.46 | 25.52 | 25.38 | 25.29 |
| CROPPING | 22.73 | 21.11 | 23.08 | 24.09 | 22.42 | 23.93 | 28.69 | 27.51 | 27.70 |
| GLOBAL-LOCAL CLIP | 24.88 | 23.61 | 24.66 | 26.16 | 24.90 | 25.83 | 31.11 | 30.96 | 30.69 |
| GLOBAL-LOCAL CLIP (SAM) | 23.68 | 25.79 | 20.01 | 26.02 | 28.62 | 21.31 | 27.04 | 25.35 | 26.40 |
| SAM-CLIP | 25.23 | 25.86 | 24.75 | 25.64 | 27.76 | 26.06 | 33.75 | 34.80 | 33.65 |
| **REF-DIFF+** | **35.16** | **37.44** | **34.50** | **35.56** | **38.66** | **31.40** | **38.62** | **37.50** | **37.82** |

Table 2: **The mIoU comparison.** The improvement is statistically significant with $p < 0.01$ under $t$-test. † denotes weakly-supervised method trained on datasets.

| METHODS | REFCOCO | | | REFCOCO+ | | | REFCOCOG | | |
|---|---|---|---|---|---|---|---|---|---|
| | VAL | TESTA | TESTB | VAL | TESTA | TESTB | VALU | TESTU | VALG |
| *Without Pre-trained Segmentor* | | | | | | | | | |
| TSEG† | **25.95** | - | - | 22.62 | - | - | 23.41 | - | - |
| **REF-DIFF** | 23.06 | **20.05** | **23.17** | 23.91 | 20.53 | 23.64 | 27.03 | 25.95 | 26.45 |
| *With Pre-trained Segmentor* | | | | | | | | | |
| REGION TOKEN | 23.43 | 22.07 | 24.62 | 24.51 | 22.64 | 25.37 | 27.57 | 27.34 | 27.69 |
| CROPPING | 24.83 | 22.58 | 25.72 | 26.33 | 24.06 | 26.46 | 31.88 | 30.94 | 31.06 |
| GLOBAL-LOCAL CLIP | 26.20 | 24.94 | 26.56 | 27.80 | 25.64 | 27.84 | 33.52 | 33.67 | 33.61 |
| GLOBAL-LOCAL CLIP (SAM) | 31.38 | 33.31 | 27.30 | 34.46 | 35.00 | 29.38 | 38.56 | 33.49 | 33.88 |
| SAM-CLIP | 26.33 | 25.82 | 26.40 | 25.70 | 28.02 | 26.84 | 38.75 | 38.91 | 38.27 |
| **REF-DIFF+** | **37.21** | **38.40** | **37.19** | **37.29** | **40.51** | **33.01** | **44.02** | **44.51** | **44.26** |

The mIoU and oIoU comparisons are shown in Tables 1 and 2, respectively. We can observe that both REF-DIFF and REF-DIFF+ exhibit significantly superior performance in comparison with the related methods and baselines. For setting a), our REF-DIFF outperforms SOTA wealy-supervised model (*i.e.*, TSEG) significantly on REFCOCO+ and REFCOCOG, even without any pre-trained segmentor (*e.g.*, SAM) and training data. For setting b), benefiting from the combination of both generative and discriminative models, our REF-DIFF+ achieves an improvement of approximately 10 oIoU and mIoU on REFCOCO, REFCOCO+, and REFCOCOG in comparison with SOTA methods. From

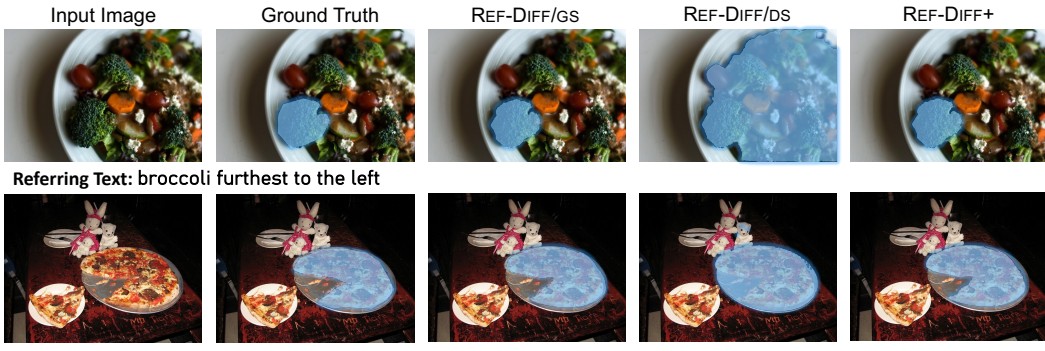

Figure 3: **Effectiveness of generative model in localization capability.** The discriminative model focuses more on whether the image contains text-related content, which results in mistakenly selecting larger regions.

the comparison between GLOBAL-LOCAL CLIP (SAM) or SAM-CLIP with other methods, only the improvement of mIoU can be observed. We analyze that segmentation from SAM is highly detailed (*e.g.*, it may generate many proposals overlapped with the same object, especially for small objects), which easily increases the possibility of CLIP filtering out erroneous proposals by only using the discriminative model CLIP directly. Therefore, the improvement on oIoU is not obvious. By incorporating the generative model, the filtering of erroneous proposals is further mitigated, contributing to our superior performance. This also demonstrates that our improvement over baseline models from a deeper understanding facilitated by considering the strengths of both our generative and discriminative models. For results on PHRASECUT, kindly refer to Section H in the appendix.

## 4.3 ABLATION STUDY

We conduct an ablation study in Table 3 and Table 4, which validate the effectiveness of different components. Notably, our REF-DIFF outperforms the weakly supervised model TSEG (Strudel et al., 2022) on REFCOCO+ and REFCOCOG when using the generative model alone. When combined with the segmentor, REF-DIFF/GS consistently exhibits superior performance across all test sets to the weakly supervised model. These observations collectively show that the generative model can not only perform proposal generation but also benefit the RIS task. More analyses can be found in Section A, B, C, and D in the appendix.

Table 3: **The oIoU comparison of our REF-DIFF with different components.**

| METHODS | SEGMENTOR | GENERATIVE | DISCRIMINATIVE | REFCOCO | REFCOCO+ | REFCOCOG |
|---|---|---|---|---|---|---|
| REF-DIFF | | ✓ | | 16.12 | 17.50 | 20.39 |
| REF-DIFF/GS | ✓ | ✓ | | 26.04 | 26.68 | 26.84 |
| REF-DIFF/DS | ✓ | | ✓ | 33.64 | 34.43 | 34.83 |
| REF-DIFF+ | ✓ | ✓ | ✓ | **35.16** | **35.56** | **38.62** |

Table 4: **The mIoU comparison of our REF-DIFF with different components.**

| METHODS | SEGMENTOR | GENERATIVE | DISCRIMINATIVE | REFCOCO | REFCOCO+ | REFCOCOG |
|---|---|---|---|---|---|---|
| *Weakly-supervised Method* | | | | | | |
| TSEG | | | | 25.95 | 22.62 | 23.41 |
| *Zero-shot Method* | | | | | | |
| REF-DIFF | | ✓ | | 22.62 | 22.72 | 27.03 |
| REF-DIFF/GS | ✓ | ✓ | | 29.82 | 30.06 | 30.73 |
| REF-DIFF/DS | ✓ | | ✓ | 35.27 | 35.72 | 41.84 |
| REF-DIFF+ | ✓ | ✓ | ✓ | **37.21** | **37.29** | **44.02** |

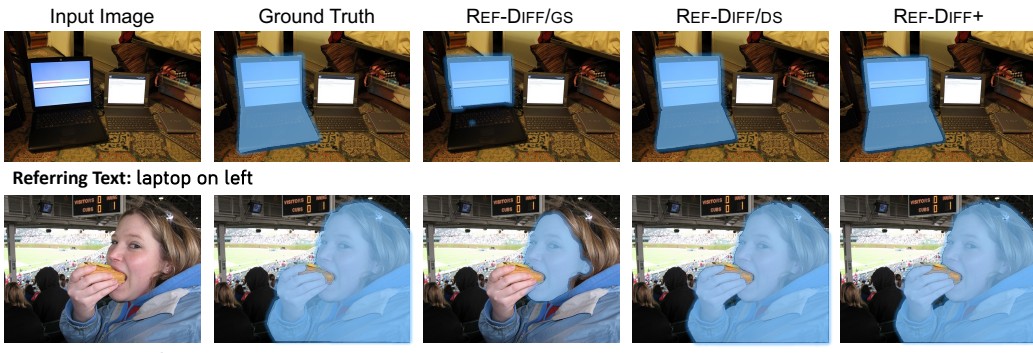

| Input Image | Ground Truth | REF-DIFF/GS | REF-DIFF/DS | REF-DIFF+ |

Figure 4: **Effectiveness of discriminative model.** The generative model exhibits higher sensitivity to salient visual features, which can result in partial segmentation when solely relying on the generative model. By integrating the discriminative model, we can effectively mitigate such errors and achieve more accurate results.

## 4.4 EFFECT OF GENERATIVE MODEL

**Segmentation Capability.** From Figure 2 it can be observed that even without the assistance of the segmentor and CLIP, our REF-DIFF is able to accurately segment the corresponding content. This is primarily attributed to the attention projection of the generative model onto the relevant visual content (kindly refer to Sec. 4.6). In the second example, despite the presence of two instances of the same object (person) in the image, our REF-DIFF is still capable of performing reasonably accurate segmentation. This highlights the significant potential of the generative model, as it exhibits 1) segmentation capabilities comparable to segmentation models and 2) categorization capabilities comparable to discriminative models. For visualizations of proposals generated by attention map, kindly refer to Section E in the appendix.

**Localization Capability.** From the first example shown in Figure 3, we can see that the discriminative model fails to accurately locate the leftmost broccoli. Similarly, in the second example, CLIP fails to eliminate the redundant region of the plate. We analyze this issue may arise due to the inherent limitation of the discriminative model, which is trained to identify whether the given text and image are well aligned. So it lacks the capability of localizing objects within the image. Consequently, relying solely on the discriminative model to discern whether a region contains redundant content becomes challenging. However, when we combine the generative and discriminative models, we are able to achieve the best results. For quantitative analysis of the generative model, kindly refer to Section F.1 in the appendix.

## 4.5 EFFECT OF DISCRIMINATIVE MODEL

In the first example depicted in Figure 4, we observe that the generative model incorrectly segments the screen as a separate object due to its prominence as a significant visual feature of a laptop, attracting more attention from the generative model. Similarly, in the second example, the generative model places greater emphasis on the person's face, resulting in incomplete segmentation. However, these errors can be effectively mitigated in our full model REF-DIFF, due to the robust categorization capability of the discriminative model. For quantitative analysis of the discriminative model, kindly refer to Section F.2 in the appendix.

## 4.6 ATTENTION FROM GENERATIVE MODEL

To investigate the region that the generative model focuses on, we provide visualizations of sample images along with the attention weights assigned to each token in the generative model, as depicted in Figure 5. We find that: 1) Generative models exhibit contextual understanding capabilities, as they effectively allocate attention to the relevant regions in the image, thereby accomplishing both localization and segmentation tasks. Notably, the attention assigned to the subject token closely aligns with the final segmentation results. This finding elucidates why generative models can perform well in Zero-shot referring segmentation tasks without relying on a separate segmentor or a discriminative model. 2) In contrast to classification models, where the first token often represents

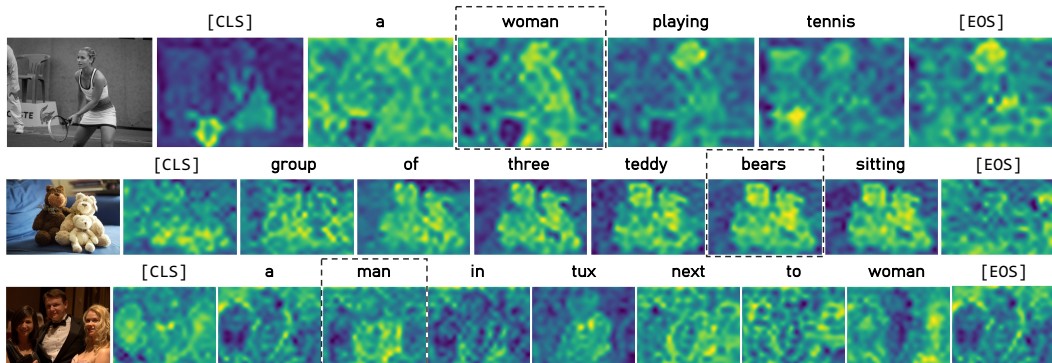

Figure 5: **Attention from the generative model.** Generative model projects attention to different regions of the image based on different tokens, which is the key reason for the effectiveness of REF-DIFF. The dashed box highlights the root token and its corresponding attention map.

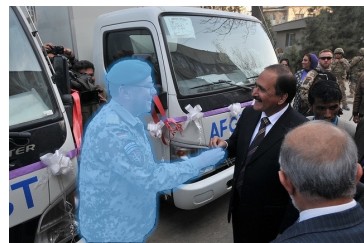
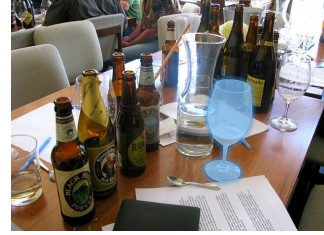
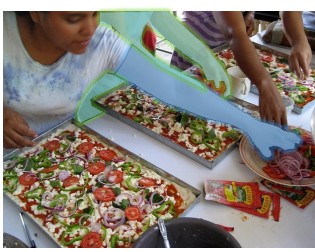

Military guy green beret

Wine glass in front next to water carafe

Second arm from left

Figure 6: **Case studies.** Blue indicates the predicted regions. The third case is a failure example, where green denotes ground-truth regions. REF-DIFF demonstrates its ability to accurately segment objects based on the provided referring texts, even in the presence of complex spatial relationships within the image.

the entirety of the text, we observe that the information associated with the first token in our generative model does not capture the complete context. However, as discussed earlier, the subject token successfully captures the comprehensive information from the text, enabling accurate results. For discussion on different attention tokens, kindly refer to Section G in the appendix.

## 4.7 CASE STUDY

We presented three examples of varying difficulty in Figure 6 to showcase the effectiveness of our REF-DIFF. In the first example, we observed that REF-DIFF demonstrates the capability to accurately identify and segment the correct object within similar objects. In the second example, despite the presence of numerous objects in the image and the complex spatial relationships, REF-DIFF successfully identifies the correct objects through the accurate understanding of the generative and discriminative models. In the final example, we encountered a segmentation failure of REF-DIFF due to the presence of some degree of ambiguity in the referring expression. REF-DIFF incorrectly identifies the leftmost hand as the first arm, resulting in a segmentation error. Enhancing the robustness of REF-DIFF is an area of future work that deserves further investigation. For a discussion of broader impact and limitations, kindly refer to Section I in the appendix.

## 5 CONCLUSION

In this work, we proposed a novel Referring Diffusional segmentor (REF-DIFF) for zero-shot referring image segmentation, which effectively leverages the fine-grained multi-modal information from generative models. We demonstrated that REF-DIFF, using a generative model alone, can outperform SOTA weakly-supervised models without requiring a proposal generator in most popular datasets. Moreover, by combining both generative and discriminative models, REF-DIFF+ outperformed these competing methods by a significant margin. Overall, our work presented a simple yet promising direction for zero-shot referring image segmentation by exploiting the potential of generative models, which brought new perspectives for addressing the challenges of this task.

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

This appendix mainly contains:

- Effect of image resolution for the generative model in Section A

- Analyses of positional bias for the discriminative model in Section B

- Analyses of $\alpha$ for combining generative and discriminative models in Section C

- Analyses of inference time complexity in Section D

- Visualization of proposals from the generative model in Section E

- Exploration in the strength of generative and discriminative models in Section F

- Discussion on attention token in Section G

- Further results on PHRASECUT in Section H

- Discussion on broader impact and limitations in Section I

## A  EFFECT OF IMAGE RESOLUTION FOR THE GENERATIVE MODEL

Table 5: **Comparison of variants with different generative resolutions on oIoU.**

| METHODS | GENERATIVE RESOLUTION | REFCOCO | REFCOCO+ | REFCOCOG |
|---------|----------------------|---------|----------|----------|
| REF-DIFF | $512 \times 512$ | 14.53 | 15.76 | 18.30 |
| REF-DIFF | $768 \times 768$ | 14.66 | 16.04 | 19.88 |
| REF-DIFF | $1024 \times 1024$ | **16.12** | **17.50** | **20.39** |

The generative model is capable of generating images with different resolutions. Generally, the higher the resolution of the generated images, the more fine-grained and detailed structures they will exhibit. To investigate the effect of image resolution bringing on this task, we conduct variants on REF-DIFF that exclusively employ the generative model. This allows us to focus on exploring the generative model while eliminating the influence of other factors, *e.g.*, the external segmentor and the discriminative model. Specifically, we consider three resolutions, *i.e.*, $512 \times 512$, $768 \times 768$, and $1024 \times 1024$. As indicated in Table 5, the performance of three variants exhibits a positive correlation with the image resolution. We analyze that higher resolutions contribute to more refined attention maps, resulting in more accurate attention projections on different regions.

## B  ANALYSES OF POSITIONAL BIAS FOR THE DISCRIMINATIVE MODEL

Table 6: **Proportion of positional token in referring texts.**

| REFCOCO | REFCOCO+ | REFCOCOG |
|---------|----------|----------|
| 47.2% | <0.1% | 16.2% |

Table 7: **The ablation study of positional bias on oIoU.**

| METHODS | POSITIONAL BIAS | REFCOCO | REFCOCO+ | REFCOCOG |
|---------|-----------------|---------|----------|----------|
| REF-DIFF/DS |  | 32.85 | **34.43** | 34.59 |
| REF-DIFF/DS | ✓ | **33.64** | **34.43** | **34.82** |

These types of positional tokens (*i.e.*, left, right, top, and bottom) usually exist in the referring text. They can provide explicit location descriptions for RIS but remain unexplored in this task. In Table 6, we present the proportion of referring texts that contain explicit location descriptions. It shows that REFCOCO has nearly 47.2% referring texts that possess this feature. By incorporating these positional bias, our REF-DIFF/DS demonstrates a noticeable improvement on REFCOCO (33.64 *v.s.* 32.85 in Table 7). A similar positive effect is observed in the comparison on REFCOCOG. This indicates that the positional bias can be used to enhance the RIS performance on datasets that include positional descriptions.

Table 8: **The ablation study of different $\alpha$ on oIoU.**

| METHODS | $\alpha$ | REFCOCO | REFCOCO+ | REFCOCOG |
|---|---|---|---|---|
| REF-DIFF/DS | 0.00 | 33.64 | 34.43 | 34.83 |
| REF-DIFF+ | 0.05 | 35.12 | 34.00 | 37.87 |
| REF-DIFF+ | 0.10 | **35.16** | **35.56** | **38.62** |
| REF-DIFF+ | 0.20 | 34.50 | 35.16 | 37.63 |
| REF-DIFF+ | 0.50 | 29.79 | 31.45 | 34.12 |
| REF-DIFF/GS | 1.00 | 26.04 | 26.68 | 26.84 |

## C ANALYSES OF $\alpha$ FOR COMBINING GENERATIVE AND DISCRIMINATIVE MODELS

We note that the hyperparameter $\alpha$, which determines the balance between discriminative and generative models, is crucial and directly affects the model's final performance. As shown in Table 8, we find that the model achieves optimal performance when $\alpha$ is set to $0.1$, while performance tends to deteriorate when $\alpha$ is either lower or larger. These results highlight the critical role of $\alpha$ in achieving the desired performance in the model.

## D TIME COMPLEXITY

We report the inference time of different variants of REF-DIFF in Table 9. We can see that when using more modules, REF-DIFF requires more time for inference. Since we only need a small number of DDIM steps for Stable Diffusion, the time consumption of the generative model seems also acceptable.

Table 9: **Inference time cost on different variants.**

| METHODS | SEGMENTOR | GENERATIVE | DISCRIMINATIVE | INFERENCE TIME | MULTIPLE |
|---|---|---|---|---|---|
| REF-DIFF | | ✓ | | 1.17s | 1.00× |
| REF-DIFF/GS | ✓ | ✓ | | 2.74s | 2.34× |
| REF-DIFF/DS | ✓ | | ✓ | 2.64s | 2.26× |
| REF-DIFF+ | ✓ | ✓ | ✓ | 3.18s | 2.72× |

## E VISUALIZATION OF PROPOSALS FROM GENERATIVE MODEL

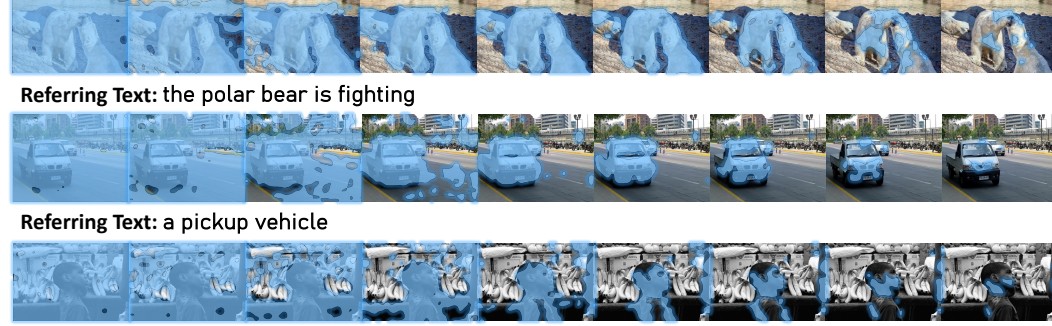

**Referring Text:** the polar bear is fighting

**Referring Text:** a pickup vehicle

**Referring Text:** black boy standing beside bunch of bananas at the market

Figure 7: **Visualization of proposals with different threshold in Equation 7.**

Through the extraction of proposals from the attention information of the generative model, our proposed REF-DIFF demonstrates the capability to perform the RIS task without relying on an external segmentor. Figure 7 demonstrates the proposals extracted by the generative model. In contrast to the approach involving an external segmentor, the generative model, with its language conditioning information, selectively extracts proposals that are directly relevant to the referring text, thereby

avoiding the inclusion of numerous irrelevant objects present in the image. This proposal extraction approach enhances the efficiency and relevance of the RIS task within our REF-DIFF.

## F  STRENGTH OF THE GENERATIVE MODEL AND DISCRIMINATIVE MODEL

### F.1  STRENGTH OF THE GENERATIVE MODEL

We have organized quantitative experiments on oIoU to explore the advantages of the generative model. We divided REFCOCO into several subsets based on the length of the referring text and calculated their performance separately. Here, REF-DIFF refers to the complete network that includes the segmentor, generative model, and discriminative model; while REF-DIFF/DS removes the generative model.

Table 10: **Results with different lengths of referring text.**

| METHOD | 1-2 | 3-4 | 5-6 | 7-8 | 9+ |
|---|---|---|---|---|---|
| REF-D/DS | 42.14 | 38.64 | 32.93 | 28.89 | 24.36 |
| REF-DIFF+ | 43.93(+4.2%) | 38.27(+1.0%) | 34.29(+4.1%) | 33.28(+15.2%) | 28.47(+16.9%) |

As shown in Table 10, we can observe that REF-DIFF integrating the generative model is more pronounced for longer referring text. This is because the generative model can provide information that the discriminative model struggles to capture in complex referring relationships. Therefore, the advantage of the generative model can be attributed to its stronger localization capability.

### F.2  STRENGTH OF THE DISCRIMINATIVE MODEL

We also quantified the inner area proportion, specifically, the ratio of the predicted segmentation lies within the ground truth result in Table 11. We found that the drawback of the generative model is its heightened sensitivity to salient visual features, which makes it prone to producing partial segmentation results. This is the strength of the discriminative model.

Table 11: **Proportion of inner area proportion.**

| METHOD | PROPORTION OF AREA |
|---|---|
| REF-DIFF/GS | 27.8% |
| REF-DIFF+ | 41.2% |

## G  DISCUSSION ON ATTENTION TOKEN

Table 12: **The ablation study of different attention tokens on oIoU.**

| METHODS | TOKEN | REFCOCO | REFCOCO+ | REFCOCOG |
|---|---|---|---|---|
| REF-DIFF/GS | CLS | 17.55 | 18.40 | 16.39 |
| REF-DIFF/GS | END | 15.03 | 16.82 | 16.76 |
| REF-DIFF/GS | ROOT | **26.04** | **26.68** | **26.84** |

We validate whether commonly used sentence representations corresponding to tokens and end-of-sentence tokens effectively model referring information for the generative model. As shown in Table 12, we find that compared to the root token mentioned in this paper, commonly used `[CLS]` and end-of-sentence tokens achieved lower performance. This also supports our finding that for the generative model, the root token of the referring text contains more comprehensive referential information than the other tokens, making it more suitable for referring segmentation tasks.

## H    RESULTS ON PHRASECUT

Furthermore, we conduct additional experiments on PHRASECUT dataset in Table 13. We can see that our REF-DIFF also outperforms the competing method GLOBAL-LOCAL by a large margin on oIoU. This further validates the great generalization of our REF-DIFF on other datasets.

Table 13: **Results on PRASECUT.**

| METHODS | oIoU | mIoU |
|---|---|---|
| *Without Pre-trained Segmentor* | | |
| TSEG | - | 30.12 |
| **REF-DIFF** | **12.79** | **31.80** |
| *With Pre-trained Segmentor* | | |
| GLOBAL-LOCAL CLIP | 23.64 | - |
| **REF-DIFF+** | **29.42** | **41.75** |

## I    BROADER IMPACT AND LIMITATIONS

Zero-shot referring image segmentation has broad applications in industrial and real-world domains, such as image editing, robot control, and human-machine interaction. our REF-DIFF, through the combination of generative and discriminative models, has successfully demonstrated the feasibility of Training-free yet high-quality referring segmentation in various data-scarce scenarios. This significantly reduces the deployment cost of artificial intelligence in related fields. However, due to the existence of pre-trained modules, the inference stage still incurs high computational overhead. Moreover, referring expression texts are sensitive to ambiguity (see Sec. 4.7), which currently results in noticeable segmentation errors. In the future, we will further investigate zero-shot referring segmentation with lower computational costs and higher robustness.

