# OpenReview forum: "Ref-Diff: Zero-shot Referring Image Segmentation with Generative Models"
_ICLR.cc/2024/Conference — ICLR 2024 Conference Withdrawn Submission_

### Official Review · Reviewer_YFJv · 2023-10-29

**Soundness:** 3 good
**Presentation:** 3 good
**Contribution:** 2 fair
**Rating:** 3
**Confidence:** 4

**Summary:**

This paper studies an important problem: zero-shot referring segmentation. It proposes to combine generative model and discriminative model in a unified framework. Depending on whether SAM is used, two versions of the proposed method are REF-DIFF and REF-DIFF+. In general, this is an interesting work. However, there are several issues that need to be addressed.

**Strengths:**

1. The paper is well-written and easy to follow. The figures are clear and looks good.
2. Good performance is achieved on different datasets.

**Weaknesses:**

1. how to generate the mask proposals using a pre-trained segmentor? What are the input and output? The paper only mentioned that pre-trained segmentor is SAM. But it is unclear to me how the SAM is used. In my experience, how the SAM is used could have big impact on the segmentation masks.
2. In my opinion, ref-diff+ is a mixure of stable diffussion, CLIP, and SAM. I do not see novel stuff/insights in the proposed method. It is questionable whether combing different methods and doing engineering job to make it work are enough for publication in this conference?
3. How many steps are used in generative process? What is the impact of the number of the steps?
4. What are the details of gaussian noise function in Eq 3? Are there hyper-parameters to control the noise, i.e., mean and variance? If so, what is the impact of these hyper-parameters?
5. In table 1, there are no results for TSEG. What is the point of putting it in the talbe? It is better to remove it.
6. In table 1, the comparisons between REF-DIFF+ with methods (REGION TOKEN, CROPPING, and GLOBAL-LOCAL CLIP) are unfair because these methods do not use SAM. As we know, SAM was trained on millions of images, costing millions of dollars. The masks generated by SAM should provide large benefits to the proposed method. Also, it is strange that GLOBAL-LOCAL CLIP (SAM) is worse than GLOBAL-LOCAL CLIP, meaning that SAM is useless for GLOBAL-LOCAL CLIP. Any chance to make it work? What are the details for SAM-CLIP?

**Questions:**

See weaknesses

---

### Official Review · Reviewer_83NR · 2023-10-30

**Soundness:** 1 poor
**Presentation:** 2 fair
**Contribution:** 1 poor
**Rating:** 3
**Confidence:** 5

**Summary:**

The authors propose a zero-shot referring segmentation method by leveraging diffusion model. Utilizing the cross attention values from target tokens, the proposed method can extract segmentation mask out of mask candidates. A multi-modal generative model has been used both in generative and discriminative processes.

**Strengths:**

- Generative model is used for zero-shot segmentation.
- The paper is easy to read.

**Weaknesses:**

- At first glance, it may appear to be a zero-shot method but I do not agree with that. Stable diffusion is trained with numerous images and captions, which makes it hard to assume the proposed method as zero-shot. This also means that if an object which is out of vocabulary from CLIP's tokenizer will crash this method. Also, the existence of pre-trained segmentor disrupts the contribution of this paper.
- The explanation about positional bias is not clear. I don't know whether it comes from literal direction words such as 'left', 'right' or something else. If the author is proposing something in the paper, the definition of derivation must exist.
- Similarly, I don't understand how the 'ROOT' token is chosen. It is mentioned that they used syntax analysis but I think this should be explained more in details.
- The methods in the experiment session are not reasonable. There are several other works which perform much better than the score in this paper and they are all neglected (e.x)[1].
- The concept of this method is extremely simple and straightforward. Lots of research works have leveraged attention mechanism and it is not something new. Simply generating attention masks and picking the best one does not seems to give a lesson to the community. More novelty is required in this paper.

[1] Lee, Jungbeom, et al. "Weakly Supervised Referring Image Segmentation with Intra-Chunk and Inter-Chunk Consistency." Proceedings of the IEEE/CVF International Conference on Computer Vision. 2023.

**Questions:**

- I would be appreciated if the concerns above is resolved.

---

### Official Review · Reviewer_rW9R · 2023-10-31

**Soundness:** 2 fair
**Presentation:** 2 fair
**Contribution:** 1 poor
**Rating:** 3
**Confidence:** 4

**Summary:**

The authors use an off-the-shelf diffusion model to perform zero-shot referring segmentation. They use a pre-trained segmenter (SAM) and a CLIP model to further improve their setup. They present results on referring segmentation datasets and obtain good results.

**Strengths:**

The method is training free and utilized the strong representations in text-to-image diffusion models to perform a task they were not trained on.

**Weaknesses:**

**Overview**

Using cross-modal similarity map from attention layers is explored in numerous CLIP-based segmentation settings [1-6] and simply applying same to the case of diffusion models is minimally novel. Weight-free mask proposals from Stable Diffusion is done in prior work [10] and other contributions (root token, positional bias) are highly unclear. The paper in current form has numerous flaws; two main discussed below.

**Missing method details**

* (Sec 3.2) What is the tokenization used in language encoder? BPE like CLIP? Doesn't this split individual words in sub-parts (in some cases)?
Using given example, what happens if "horse" is split into "ho", "rse" tokens? Please provide these details and explain the root token selection clearly taking into account nature of tokenization scheme.
* (Sec 3.3) "Specifically, if the text, after syntactic analysis, contains explicit direction clues, P will be a soft mask with values ranging from 1 to 0 along the given direction axis." - how is direction obtained from the text?
* Missing ablation on positional bias

**Missing Related Work**
* Missing related work on weakly supervised open-vocab segmentation. Please discuss these works (e.g. 1-6) that can perform similar task and identical settings. Also consider comparing to one of these works (e.g. row in Table 1, 2) using their pre-trained model.
* Cite and discuss prior work that uses Stable Diffusion for zero-shot referring segmentation [7, 9] (exact same task). Also compare to these baselines.
* Discuss prior work exploring discriminative abilities of Stable Diffusion.


1. Zhang, Yabo et al. “Associating Spatially-Consistent Grouping with Text-supervised Semantic Segmentation.”
2. Luo, Huaishao et al. “SegCLIP: Patch Aggregation with Learnable Centers for Open-Vocabulary Semantic Segmentation.” (ICML 2022)
3. Ranasinghe, Kanchana et al. “Perceptual Grouping in Contrastive Vision-Language Models.” (ICCV 2023).
4. Xu, Jilan et al. “Learning Open-Vocabulary Semantic Segmentation Models From Natural Language Supervision.” (CVPR 2023)
5. Cha, Junbum et al. “Learning to Generate Text-Grounded Mask for Open-World Semantic Segmentation from Only Image-Text Pairs.” (CVPR 2023)
6. Mukhoti, Jishnu et al. “Open Vocabulary Semantic Segmentation with Patch Aligned Contrastive Learning.” (CVPR 2023)

7. Burgert, Ryan et al. “Peekaboo: Text to Image Diffusion Models are Zero-Shot Segmentors.” (CVPR 2023).
8. Li, Alexander C. et al. “Your Diffusion Model is Secretly a Zero-Shot Classifier.” (ICCV 2023)
9. Karazija, Laurynas et al. “Diffusion Models for Zero-Shot Open-Vocabulary Segmentation.”
10. Tian, Junjiao et al. “Diffuse, Attend, and Segment: Unsupervised Zero-Shot Segmentation using Stable Diffusion.”

**Questions:**

Please address issues under weaknesses

1. What is inference compute requirement and latency?
2. Can you provide results on a more complex RefSeg dataset?

---

### Official Review · Reviewer_2K1U · 2023-10-31

**Soundness:** 3 good
**Presentation:** 3 good
**Contribution:** 2 fair
**Rating:** 6
**Confidence:** 4

**Summary:**

This paper observes that generative models can discern relationships between various visual elements and text descriptions which is an area yet to be explored in this task. So the authors use stable diffusion to sample from a noised image latent feature and use the attention map of ROOT token embedding (extracted with syntax analysis) and image feature as the seed of the mask. Then, the attention map is refined or selected with thresholds (Weight-free Proposal Filter in the paper) or SAM (pre-trained segmentor in the paper) together with the similarity between text embedding and map (Generative Matching in the paper).
The authors also proposed the Discriminative version parallelly. Here, they take direction bias hidden in text descriptions into consideration, which is ignored in previous works as they claimed.
Finally, they combine these two methods together and perform SOTA both in week-supervised referring segmentation and referring segmentation with pre-trained segmentors.

**Strengths:**

1. Ref-Diff is the first work to utilize the attention map in generative model in referring segmentation.
2. Ref-Diff makes use of the position bias hidden in the text which is ignored in previous works.
3. Ref-Diff achieves SOTA in zero-shot referring segmentation.

**Weaknesses:**

1. Missing reference, [1-2] also propose to use the attention map of the diffusion model to localize the objects in a weakly supervised manner, which should be cited and compared. [3] propose to use image-text pairs to augment referring segmentation, which should be cited and compared.
2. Using the text-image correlation in the diffusion model to localize objects has been widely used in published works in 2023, the novelty of the proposed Ref-Diff is limited.

[1] Xiao, Changming, et al. "From Text to Mask: Localizing Entities Using the Attention of Text-to-Image Diffusion Models." arXiv preprint arXiv:2309.04109 (2023).

[2] Zhao, Yuzhong, et al. "Generative prompt model for weakly supervised object localization." Proceedings of the IEEE/CVF International Conference on Computer Vision. 2023.

[3] Yucheng Suo, et al. "Text Augmented Spatial-aware Zero-shot Referring Image Segmentation." EMNLP, 2023

**Questions:**

see weakness